# Decomposing The Dark Matter of Sparse Autoencoders

## Abstract

Sparse autoencoders (SAEs) are a promising technique for decomposing language model activations into interpretable linear features. However, current SAEs fall short of completely explaining model performance, resulting in "dark matter"— unexplained variance in activations. In this work, we predict and verify that much of SAE dark matter can be linearly predicted from the activation vector. We exploit this fact to deconstruct dark matter into three top-level components: 1) unlearned linear features, 2) unlearned dense features, and 3) nonlinear errors introduced by the SAE. Through a scaling laws analysis, we estimate that nonlinear SAE errors stay constant as SAEs scale and serve as a lower bound of SAE performance on both an average and per-token level. We next empirically analyze the nonlinear SAE error term and show that it is not entirely a sparse sum of unlearned linear features, but that it is still responsible for some of the downstream reduction in cross entropy loss when SAE activations are inserted back into the model. Finally, we examine two methods to reduce nonlinear error: inference time gradient pursuit, which leads to a very slight decrease in nonlinear error, and linear transformations from earlier layer SAE dictionaries, which leads to a larger reduction.

## 1 Introduction

The ultimate goal for ambitious mechanistic interpretability is to understand neural networks completely from the bottom up by breaking them down into programs ("circuits") and the variables ("features") that those programs operate on (Olah, 2023). One recent successful unsupervised technique for finding features in language models has been sparse autoencoders (SAEs), which learn a dictionary of one-dimensional representations that can be sparsely combined to reconstruct model hidden activations Cunningham et al. (2023); Bricken et al. (2023). However, as observed by Gao et al. (2024), the scaling behavior of SAE width (number of latents) vs. reconstruction mean squared error (MSE) is best fit by a power law with a constant error term. This is a concern for the ambitious agenda because it implies that there are components of model hidden states that are harder for SAEs to learn and which might not be eliminated by simple scaling of SAEs. Gao et al. (2024) speculate that this component of SAE error below the asymptote might best be explained by model activations having components with denser structure than simple SAE features (e.g. Gaussian noise).

In this work, we investigate the SAE error vector as an object worth study in its own right. Thus, our direction differs from the bulk of prior work that seeks to quantify SAE failures, as these mostly focus on downstream benchmarks or simple cross entropy loss (see e.g. (Gao et al., 2024; Templeton et al., 2024; Anders & Bloom, 2024)). We find that some SAE error might come, not from pre-existing dense structures in the input as Gao et al. (2024) speculates, but from noise introduced by the SAE itself. We build on this finding to propose a preliminary breakdown of SAE error (see Fig. 1) and then investigate each component in the breakdown in turn.

### 1.1 Contributions

1. To the best of our knowledge, we are the first to show that a large fraction of SAE error can be explained with a linear transformation of the input activation, and that the norm of SAE error can be accurately predicted with a linear projection of the input activation. We also provide explanations for why SAE errors have these properties.

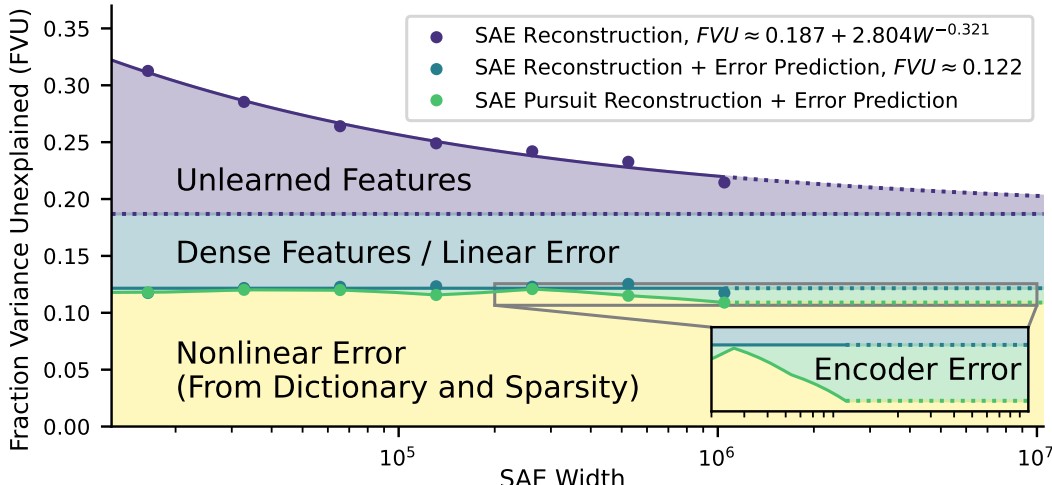

Figure 1: A breakdown of SAE dark matter. See Section 5 for how we break down the overall fraction of unexplained variance into the unlearned features, dense features/linear error, and nonlinear error. See Section 7.1 for further separating encoder error from nonlinear error.

2. We use these discovered properties of SAE error to come up with rough estimates for the magnitudes of different components of SAE error, including postulating a new type of "nonlinear error" introduced by the SAE architecture and sparsity constraint.

3. To the best of our knowledge, we are the first to examine per-token SAE scaling. We show that SAE nonlinear error serves as a per token error lower bound (in addition to serving as an overall error lower bound).

4. We investigate the nonlinear SAE error component and find that it affects downstream cross entropy loss in proportion to its norm, is harder to learn SAEs for, and is less likely to consist of unlearned linear features from the input.

5. We show that inference time gradient pursuit increases the fraction of variance explained by SAEs, but only very slightly decreases the magnitude of the nonlinear error we discovered. Additionally, we show that SAEs trained on previous components can also be used to slightly decrease nonlinear error, and indeed SAE error overall.

## 2 RELATED WORK

**Language Model Representation Structure:** The linear representation hypothesis (LRH) (Park et al., 2023; Elhage et al., 2022) claims that language model hidden states can be decomposed into a sparse sum of linear feature directions. The LRH has seen recent empirical support with *sparse autoencoders*, which have succeeded in decomposing much of the variance of language model hidden states into such a sparse sum, as well as a long line of work that has used probing and dimensionality reduction to find causal linear representations for specific concepts (Alain, 2016; Nanda et al., 2023; Marks et al., 2024; Gurnee, 2024). On the other hand, some recent work has questioned whether the linear representation hypothesis is true: Engels et al. (2024) find multidimensional circular representations in Mistral (Jiang et al., 2023) and Llama (AI@Meta, 2024), and Csordás et al. (2024) examine synthetic recurrent neural networks and find "onion-like" non-linear features not contained in a linear subspace. This has inspired recent discussion about what a true model of activation space might be: Mendel (2024) argues that the linear representation hypothesis ignores the growing body of results showing the multi-dimensional structure of SAE latents, and Smith (2024b) argues that we only have evidence for a "weak" form of the superposition hypothesis holding that only *some* features are linearly represented.

**SAE Errors and Benchmarking:** Multiple works have introduced techniques to benchmark SAEs and characterize their error: Bricken et al. (2023), Gao et al. (2024), and Templeton et al. (2024) use manual human analysis of features, automated interpretability, downstream cross entropy loss

when SAE reconstructions are inserted back into the model, and feature geometry visualizations; Karvonen et al. (2024) use the setting of board games, where the ground truth features are known, to determine what proportion of the true features SAEs learn; and Anders & Bloom (2024) use the performance of the model on NLP benchmarks when the SAE reconstruction is inserted back into the model. More specifically relevant to our main direction in this paper studying properties of the SAE reconstruction error vector, Gurnee (2024) finds that SAE reconstruction errors are *pathological*, that is, when SAE reconstructions are inserted into the model, they have a larger effect on cross entropy loss than random perturbations to the same layer equal in norm to the SAE error. Follow up work by Heimersheim & Mendel (2024) and Lee & Heimersheim (2024) find that this effect disappears when the random baseline is replaced by a perturbation in the direction of the difference between two random activations.

**SAE Scaling Laws:** Anthropic (2024), Templeton et al. (2024), and Gao et al. (2024) study how SAE MSE scales with respect to FLOPS, sparsity, and SAE width, and define scaling laws with respect to these quantities. Templeton et al. (2024) also study how specific groups of language features like chemical elements, cities, animals, and foods, and show that SAEs predictably learn these features in terms of their occurrence. Finally, Bussmann et al. (2024) find that larger SAEs learn two types of dictionary vectors not present in smaller SAEs: features not present at all in smaller SAEs, and more fine-grained "feature split" versions of features in smaller SAEs.

## 3 DEFINITIONS

In this paper, we adopt the *weak* linear hypothesis (Smith, 2024b), a generalization of the linear representation hypothesis which only holds that some features in language models are represented linearly. Formally, for hidden model activations $\mathbf{x} \in \mathbb{R}^d$, we write

$$\mathbf{x} = \sum_{i=0}^{n} \mathbf{w}_i \boldsymbol{y}_i + \texttt{Dense}(\mathbf{x}) \tag{1}$$

for linear features $\{\boldsymbol{y}_1, \ldots, \boldsymbol{y}_n\}$ and random vector $\mathbf{w} \in \mathbb{R}^n$, where $\mathbf{w}$ is sparse ($\|\mathbf{w}\|_1 \ll d$) and $\texttt{Dense}(\mathbf{x})$ is a random vector representing the dense component of $\boldsymbol{x}$. $\texttt{Dense}(\mathbf{x})$ might be Gaussian noise, nonlinear features as described by Csordás et al. (2024), or anything else not represented in a low-dimensional linear subspace.

Now consider a sparse autoencoder $\texttt{Sae} \in \mathbb{R}^d \to \mathbb{R}^d$ which seeks to minimize $\|\mathbf{x} - \texttt{Sae}(\mathbf{x})\|_2$ while using a small number of active latents. In this work, we are agnostic as to the architecture or training procedure of the sparse autoencoder; see (Bricken et al., 2023; Cunningham et al., 2023; Gao et al., 2024; Templeton et al., 2024) for such details. We now define $\texttt{SaeError}(\mathbf{x})$ such that

$$\mathbf{x} = \texttt{Sae}(\mathbf{x}) + \texttt{SaeError}(\mathbf{x}). \tag{2}$$

Now, say the SAE has $m$ latents. Since by assumption $\texttt{Dense}(\mathbf{x})$ cannot be represented in a low-dimensional linear subspace, the sparsity limited SAE will not be able to learn it. Thus, we will assume that the SAE learns only the $m$ most common features $\boldsymbol{y}_0, \ldots, \boldsymbol{y}_{m-1}$. Crucially different from the typical assumptions, however, we will also assume that the SAE introduces some error when making this approximation. We further break this new error down into that which is linearly predictable from $\mathbf{x}$, $W\mathbf{x}$ for some $W$, and that which is not, $\texttt{NonlinearError}(\mathbf{x})$. Thus we have

$$\texttt{Sae}(\mathbf{x}) = \texttt{NonlinearError}(\mathbf{x}) + W\mathbf{x} + \sum_{i=0}^{m} \mathbf{w}_i \boldsymbol{y}_i \tag{3}$$

$$\texttt{SaeError}(\mathbf{x}) = -\texttt{NonlinearError}(\mathbf{x}) - W\mathbf{x} + \texttt{Dense}(\mathbf{x}) + \sum_{i=m}^{n} \mathbf{w}_i \boldsymbol{y}_i \tag{4}$$

## 4 TESTS FOR SPLITTING SAE ERROR

### 4.1 ESTIMATING NONLINEAR ERROR

We now seek tests that will allow us to split $\texttt{SaeError}(\mathbf{x})$ into its components. The first such test consists of finding the least squares linear transformation $\boldsymbol{a}$ from $\mathbf{x}$ to $\texttt{SaeError}(\mathbf{x})$, such that

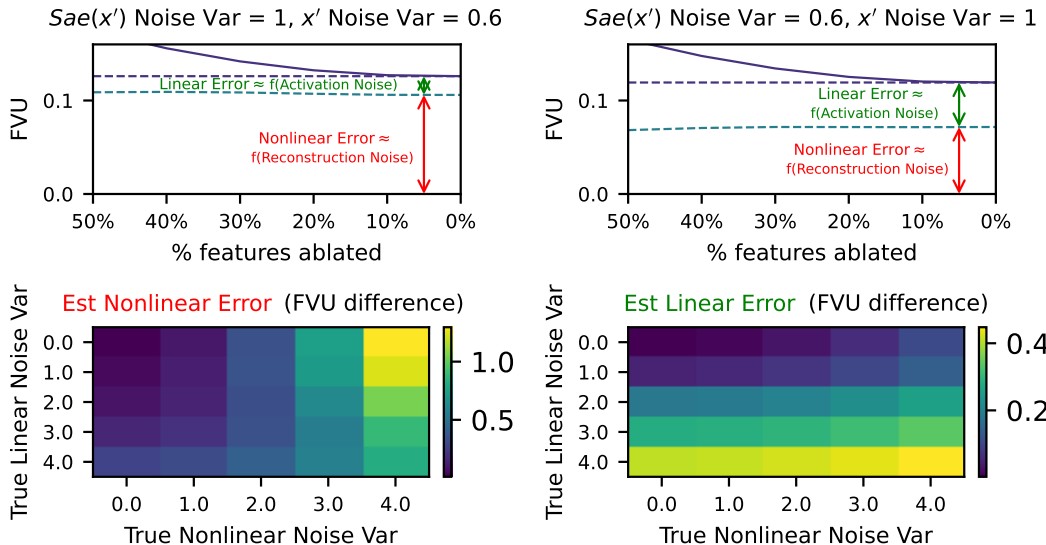

Figure 2: **Top:** When controlled amounts of noise are added to synthetic data $\texttt{Sae}(\mathbf{x}')$ and $\mathbf{x}'$, the result is a plot similar to Fig. 1. **Bottom:** The nonlinear and linear error estimates (as shown at top) accurately correlate with the amount of noise added. The exact correlation between synthetic added noise and resulting estimated error components are shown in Table 1

$\boldsymbol{a}^T\mathbf{x} \approx \texttt{SaeError}(\mathbf{x})$. The intuition behind this test is that if $-W\mathbf{x} + \texttt{Dense}(\mathbf{x}) + \sum_{i=m}^{n} \mathrm{w}_i \boldsymbol{y}_i$ is contained in a linear subspace of $\mathbf{x}$ (or equivalently if $W\mathbf{x} + \sum_{i=0}^{m} \mathrm{w}_i \boldsymbol{y}_i$ is in such a subspace), then the error of this regression exactly equals $\texttt{NonlinearError}(\mathbf{x})$. However, although such a subspace may intuitively seem likely to exist (if the $\boldsymbol{y}_i$ are all almost orthogonal, for example), its existence is not guaranteed. If such a linear transform does not exist, the percent of variance left unexplained by the regression will be an upper bound on the true variance explained by $\texttt{NonlinearError}(\mathbf{x})$. Finally, we also note that if this test is accurate, we can use it to estimate the linear component of the error, $W\mathbf{x} + \texttt{Dense}(\mathbf{x})$: the difference between the variance explained by $\texttt{Sae}(\mathbf{x})$ and the variance explained by $\mathbf{a}^T\mathbf{x}$ will approach $-W\mathbf{x} + \texttt{Dense}(\mathbf{x})$.

Thus, our ability to estimate these quantities depends on how well a linear transform to predict $\texttt{NonlinearError}(\mathbf{x})$ works on our data. Although we do not have access to the ground truth vectors $\boldsymbol{y}_i$, we *can* use a synthetic setup with a similar distribution of vectors. Specifically, given an SAE, we use the reconstruction, $\texttt{Sae}(\mathbf{x})$, as our "ground truth" activations $\mathbf{x}'$. $\mathbf{x}'$ has the useful property that it is a sparse linear sum of dictionary features (the ones that the SAE learned), and assuming the SAE was well trained, the distribution of these features and their weights should be similar to that of the true features $\boldsymbol{y}_i$.

Now that we have a ground truth $\mathbf{x}'$ that consists solely of a sparse sum of linear features, we can pass $\mathbf{x}'$ through the SAE; we find that the correct weights are recovered and the reconstruction is almost perfect: $\texttt{Sae}(\texttt{Sae}(\mathbf{x})) \approx \texttt{Sae}(\mathbf{x}) = \mathbf{x}'$. In this setting, we can now control all of the quantities we are interested in: we can simulate varying $m$ by masking SAE dictionary elements by their frequency (least frequent to most), we can simulate $\texttt{Dense}(\mathbf{x})$ by adding Gaussian noise to $\mathbf{x}'$, and we can simulate $\texttt{NonlinearError}(\mathbf{x})$ by adding Gaussian noise to $\texttt{Sae}(\mathbf{x}')$.

We run this synthetic setup with a Gemma Scope (Lieberum et al., 2024) layer 20 SAE (width $16k$, $L_0 \approx 68$). The results for different Gaussian noise amounts versus percentage of features ablated are shown in Fig. 2. We can see that, on this distribution of vectors, the test works as expected; the variance explained by $\texttt{Sae}(\mathbf{x}) + \boldsymbol{a}^T\mathbf{x}$ is a horizontal line propor-

Table 1: Correlation matrix between synthetic noise and estimated errors.

| | Estimated Linear Err | Estimated Nonlinear Err |
|---|---|---|
| $\mathbf{x}'$ Noise (True linear err) | 0.9842 | 0.1417 |
| $\texttt{Sae}(\mathbf{x}')$ Noise (True nonlinear err) | 0.0988 | 0.9036 |

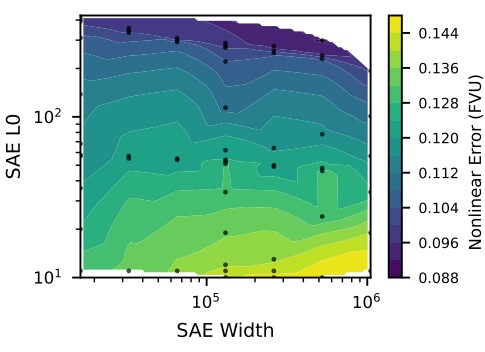 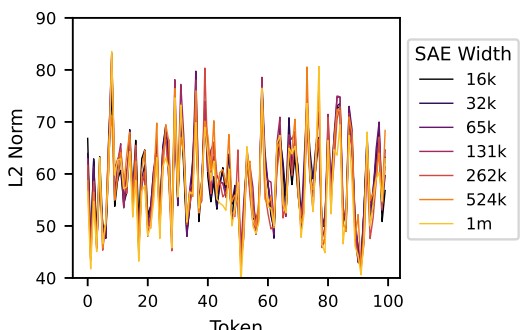

(a) Gemma 9B layer 20 fraction of total variance of **x** explained by the nonlinear error as a function of SAE L0 and SAE width, plotted in log scale as a contour plot. Larger SAE L0s have a smaller noise fraction, but noise fraction stays mostly constant with increasing SAE width.

(b) Per token norm of nonlinear errors as the size of the SAE scales. Plotted on layer 20 Gemma 9B SAEs from Gemma Scope closest to L0 = 100. The per token nonlinear error norm stays the same as the SAE gets wider; the decrease in Fig. 4 comes from the norm of the linearly predictable error decreasing.

Figure 3: `NonlinearError(x)` scaling analysis.

tional to `NonlinearError(x)`, while the gap between this horizontal line and the asymptote of the variance explained by `Sae(x)` is proportional to `Nonlinear Error`. Indeed, as shown in Table 1, these quantities are highly correlated. Thus, it seems as though at least on Gemma, this test is reasonably well supported (although note that because $\mathbf{x}'$ noise is slightly correlated with the estimated nonlinear error in Table 1, it is possible that some of the contribution to the estimated nonlinear error is from `Dense(x)`).

We also tried running this test on a sparse sum of *random* vectors, which did not work as well, possibly due to not including the structure of the SAE vectors (Giglemiani et al., 2024); see Appendix B for more details.

### 4.2 NORM PREDICTION TEST

The second test we describe aims to determine if a random vector consists mostly of a sparse sum of one-dimensional vectors. For example, we expect this to be true for **x** and `Sae(x)`, but not `NonlinearError(x)` or `Dense(x)`. First, we claim that given a vector **x**, if **x** is mostly a sparse sum of one-dimensional vectors, then there likely exists a prediction vector $\boldsymbol{a}$ such that $\boldsymbol{a}^T \mathbf{x} \approx |\mathbf{x}|_2^2$ (in other words, the norm squared of **x** can be linearly predicted from **x**). We prove that this is the case for sums of orthogonal vectors in Appendix A; the intuition is that we can set the "prediction" vector $\boldsymbol{a}$ to the sum of the vectors $\boldsymbol{y}_i$ weighted by their average value $\mathrm{w}_i$.

Similar to the first test described above, when extending to almost orthogonal vectors, we claim that this test will again be a lower bound. If the $R^2$ of the norm prediction regression is high, it is evidence that the random vector is composed of a sparse sum of vectors. Although this evidence is not conclusive (there might be other distributions with this property), it does rule out many possibilities (as for example random Gaussian noise does not have this norm prediction property).

### 5 NONLINEAR ERROR SCALING LAWS

For all experiments, unless noted otherwise, we run on $300k$ tokens of the uncopywrited subset of the Pile (Gao et al., 2020) on layer 20 of Gemma 2 9B (Team et al., 2024) using Gemma Scope (Lieberum et al., 2024) sparse autoencoders. We run with a context length of $1024$ and ignore all embeddings of tokens before token 200 in each context, as Lieberum et al. (2024) find that earlier tokens are easier for sparse autoencoders to reconstruct, and we wish to ignore the effect of token position on our results.

We run the test described in Section 4.1 on all layer 20 Gemma Scope 9B SAEs; that is, we train a linear transformation $\boldsymbol{a}$ from $\boldsymbol{x}$ to `SaeError(x)`, and use the variance left unexplained when

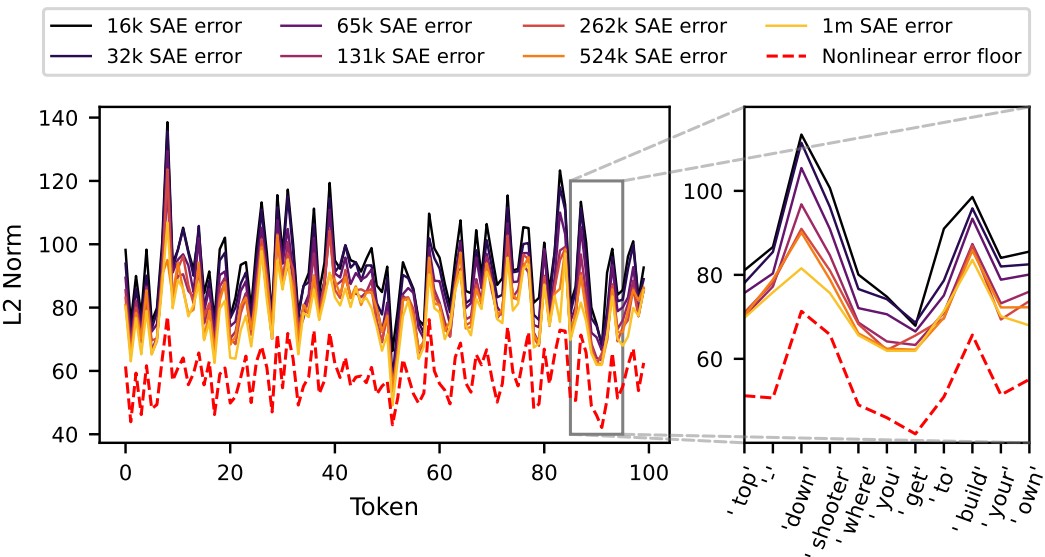

Figure 4: Per token scaling with constant nonlinear error floor, Gemma 9B SAEs from Gemma Scope closest to L0 = 60.

predicting $x$ with $\mathbf{z} = \boldsymbol{a}^T\mathbf{x} + \texttt{Sae}(\mathbf{x})$ as an estimate for $\texttt{NonlinearError}(\mathbf{x})$. As an abuse of notation, we will from now on write $\texttt{NonlinearError}(\mathbf{x})$ to mean the part of $\texttt{SaeError}(\mathbf{x})$ we cannot explain with a linear projection (which is an estimate of the true $\texttt{NonlinearError}(\mathbf{x})$ we described above). We will similarly define $\texttt{LinearError}(\mathbf{x})$ to be the part of $\texttt{SaeError}(\mathbf{x})$ we *can* explain with a linear transformation, which is approximately equal to $-W\mathbf{x} + \texttt{Dense}(\mathbf{x}) + \sum_{i=m}^{n} \mathbf{w}_i\boldsymbol{y}_i$ if our $\texttt{NonlinearError}(\mathbf{x})$ prediction is correct. Additionally, we note that we run most of our experiments in this section with $L0 \approx 60$, since this is the sparsity at which there exist Gemma Scope SAEs of matched $L0$ at different widths. Finally, for simplicity, we will drop consideration of the $W\mathbf{x}$ term, which will effectively merge it into $\texttt{Dense}(\mathbf{x})$ in our analysis.

Our main surprising finding is that at this fixed $L0 \approx 60$, the variance unexplained by $\mathbf{z}$ is approximately constant. This result is consistent with there being some component of $\texttt{NonlinearError}(\mathbf{x})$ introduced by the SAE that is not linearly explainable, assuming the distribution is similar to that described in our synthetic experiments above. We plot this $\texttt{NonlinearError}(\mathbf{x})$ as a horizontal line in Fig. 1. In this figure, we also plot the Gemma MSE vs. SAE width power law fit; it asymptotes above the horizontal $\texttt{NonlinearError}(\mathbf{x})$ line, which implies the presence of $\texttt{Dense}(\mathbf{x})$.

Another interesting result is that the $\texttt{NonlinearError}(\mathbf{x})$, although constant as SAE width scales, decreases as SAE $L_0$ increases; see Fig. 3a. We interpret this result to mean that part of the $\texttt{NonlinearError}(\mathbf{x})$ comes from the sparsity constraint in the SAE. Another interpretation might be that part of the $\texttt{NonlinearError}(\mathbf{x})$ estimate comes from noise in the input (as described in Table 1, this might be a weak yet present effect), and scaling $L_0$ allows this noise to be better fit by the SAE.

We also find that the norm of the $\texttt{NonlinearError}(\mathbf{x})$ is constant on a per token level as we scale SAE width (see Fig. 3b). This is surprising because the constant scaling behavior for $\texttt{NonlinearError}(\mathbf{x})$ described above was only on an average token level. Furthermore, the figure also shows that the nonlinear error norm floor lower bounds SAE error scaling *per token*. This is exciting because fitting a power law to just seven data points that are not averaged can be extremely noisy, so the noise floor may provide an easy way to lower bound the possible MSE able to be gained on a given token if we scaled to very large SAE width.

Finally, we note that $\texttt{NonlinearError}(\mathbf{x})$ cannot merely be explained by feature shrinkage; see Appendix C for more details.

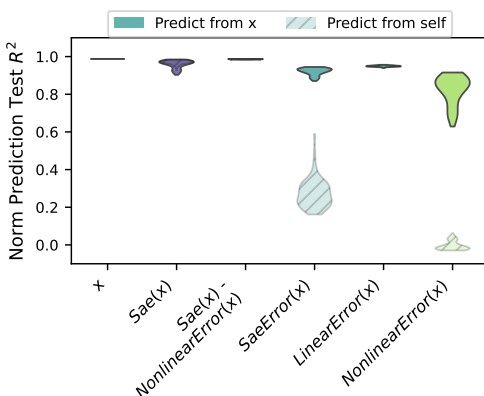 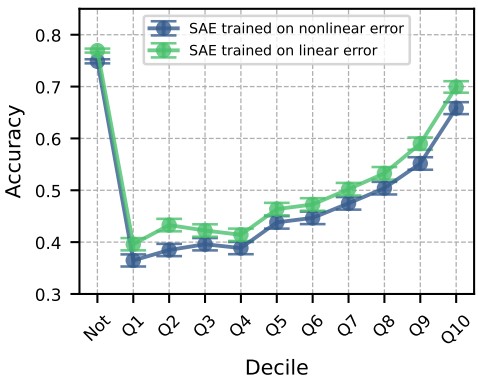

Figure 5: Violin plot of norm prediction tests for all layer 20 Gemma Scope SAEs. We plot the $R^2$ of two regressions: from each random vector to its norm squared, and from $\mathbf{x}$ to to each random vector's norm squared.

Figure 6: Auto-interpretability results on features from SAEs trained on the linear and nonlinear components of SaeError($\mathbf{x}$). "Not" means that the example feature did not activate at all, while each $Qi$ represents activating examples from decile $i$.

## 6 ANALYZING COMPONENTS OF SAE ERROR

In this section, we analyze the nonlinear and linear components to show that the split is meaningful.

### 6.1 APPLYING THE NORM PREDICTION TEST

Our first result runs the norm prediction test from Section 4.2 on six different random vectors: $\mathbf{x}$, Sae($\mathbf{x}$), Sae($\mathbf{x}$)−NonlinearError($\mathbf{x}$) (just the linearly predictable part of the SAE reconstruction), SaeError($\mathbf{x}$), LinearError($\mathbf{x}$), and NonlinearError($\mathbf{x}$). The results are shown as a violin plot for each component across all layer 20 Gemma Scope SAEs in Fig. 5 under the "Predict from self" label.

Firstly, we note that $\|\boldsymbol{x}\|_2^2$ can almost be perfectly predicted from $\boldsymbol{x}$. This is reassuring news for the linear representation hypothesis, as it implies that $\boldsymbol{x}$ can indeed well be modeled as the sum of many one-dimensional features, at least from the perspective of this test.

We also find that the two components containing NonlinearError($\mathbf{x}$), the NonlinearError($\mathbf{x}$) itself and the SaeError($\mathbf{x}$), have a very low score on this test. This supports our hypothesis that unlike $\boldsymbol{x}$, these components do not consist mostly of a sparse sum of linear features, and may be partly nonlinear error from the SAE.

One potential problem with this test is that Sae($\mathbf{x}$) − NonlinearError($\mathbf{x}$) and LinearError($\mathbf{x}$) are by construction both linear transforms of $\mathbf{x}$, so as long as these transforms are full rank, this test is equivalent to using $\mathbf{x}$ as the input to the regression (instead of just the component itself). To assuage this concern, we again run the norm prediction test but use $\mathbf{x}$ as the input to the regression instead of the target random vector itself. The idea here is that if the vector consists mostly of linear features that were present in $\mathbf{x}$, then we should be able to predict its norm. Empirically, we find that this test is more powerful than running just on the vector itself, but results in the same pattern overall; in Fig. 5, we plot these as "Predict from $\mathbf{x}$".

Finally, we note as an aside that it is interesting that $\|\texttt{SaeError}(\mathbf{x})\|_2^2$ can be almost perfectly predicted from $\mathbf{x}$, as this implies that a simple linear probe on $\mathbf{x}$ can predict the norm of SAE error with high accuracy across SAE widths and L0s.

### 6.2 TRAINING SAES ON SAEERROR($\mathbf{x}$) COMPONENTS

Another empirical test we run is training an SAE on NonlinearError($\mathbf{x}$) and LinearError($\mathbf{x}$). Our hypothesis is that it will be harder to learn an SAE for

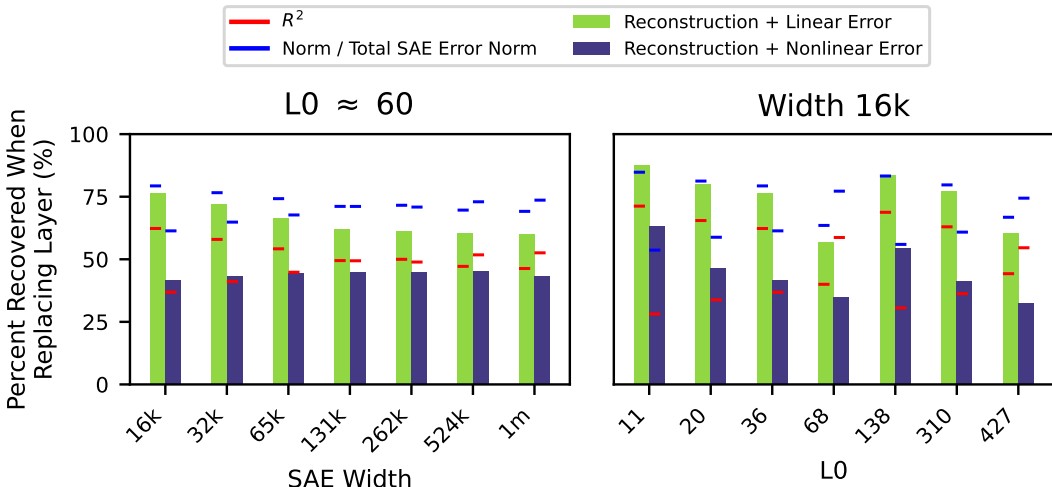

Figure 7: Results of intervening in the forward pass and replacing $\mathbf{x}$ with $\texttt{Sae}(\mathbf{x}) + \texttt{NonlinearError}(\mathbf{x})$ and $\texttt{Sae}(\mathbf{x}) + \texttt{LinearError}(\mathbf{x})$ during the forward pass. Reported in percent of cross entropy loss recovered with respect to the difference between the same intervention with $\texttt{Sae}(\mathbf{x})$ and with the normal model forward pass.

$\texttt{NonlinearError}(\mathbf{x})$ if it indeed does not consist primarily of one-dimensional SAE vectors. We choose a fixed Gemma Scope layer 20 SAE with $16k$ latents and $L_0 \approx 60$ to generate $\texttt{SaeError}(\mathbf{x})$ from. This SAE has nonlinear and linear components of the error approximately equal in norm and $R^2$ of the total $\texttt{SaeError}(\mathbf{x})$ they explain, so it presents a fair comparison. We train SAEs to convergence (about 100M tokens) on each of these components of error and find that indeed, the SAE trained on $\texttt{NonlinearError}(\mathbf{x})$ converges to a fraction of variance unexplained an absolute 5 percent higher than the SAE trained on the linear component of SAE error ($\approx 0.59$ and $\approx 0.54$ respectively).

One confounding factor is that the linear component of SAE error additionally contains $\texttt{Dense}(\mathbf{x})$, which may also be harder for the SAE to learn. Thus, we additionally examine the *interpretability* of the learned features using automated interpretability techniques first proposed by Bricken et al. (2023). Specifically, we use the implementation introduced by Juang et al. (2024), where a language model (we use Llama 3.1 70b (AI@Meta, 2024)) is given top activating examples to generate an explanation, and then must use only that explanation to predict if the feature fires on a test context. Our results, shown in Fig. 6, show that indeed, the SAE trained on linear error produces features that are about an absolute $5\%$ more interpretable on every decile of feature activation (we run on 1000 random features for both SAEs, and for each feature use 7 examples in each of the 10 feature activation deciles, as well as 50 negative examples).

### 6.3 DOWNSTREAM CROSS ENTROPY LOSS OF $\texttt{SAEERROR}(\mathbf{x})$ COMPONENTS

A common metric used to test SAEs is the percent of cross entropy loss recovered when the SAE reconstruction is inserted into the model in place of the original activation versus an ablation baseline (see e.g. Bloom (2024)). We modify this test to specifically examine the different components of $\texttt{SaeError}(\mathbf{x})$: we compare to the baseline of inserting $\texttt{Sae}(\mathbf{x})$ in place of $\mathbf{x}$, and see what percent of the cross entropy loss is recovered when replacing $\texttt{Sae}(\mathbf{x})$ with the linear and nonlinear components of $\texttt{SaeError}(\mathbf{x})$. As a baseline, we compare against what percent we would "expect" each component to recover for two notions of "expectation": the average percent of the total norm of the SAE error that the component is, and the percent of the variance that the component recovers between $\texttt{Sae}(\mathbf{x})$ and $\mathbf{x}$. The results, shown in Fig. 7, show that for the most part these are reasonable predictions for both types of error. That is, for the most part nonlinear nor linear error proportionally contribute to the SAE's increase in downstream cross entropy loss when considered by themselves, with possibly a slightly higher contribution than expected for the linear component.

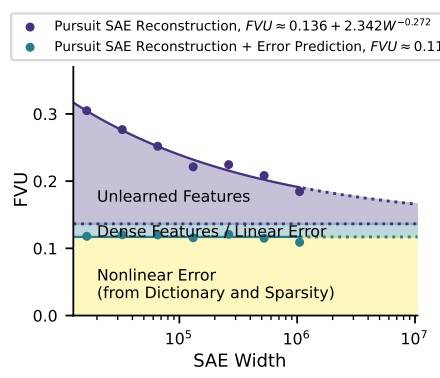
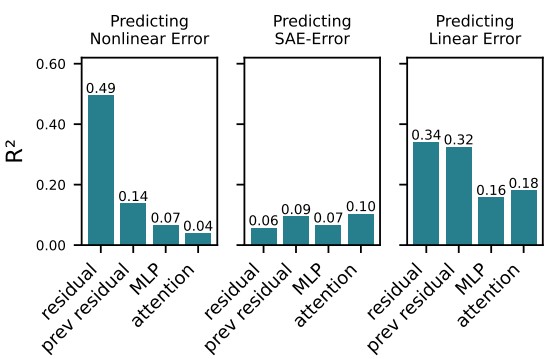

(a) SAE error breakdown vs. SAE width for inference time optimized reconstructions from Gemma Scope $L_0 \approx 60$ dictionaries.

(b) The $R^2$ of predicting parts of SAE error from the SAE reconstructions of adjacent model components, layer 20 Gemma Scope $L_0 \approx 60$, $16k$ SAE width.

Figure 8: Investigations towards reducing nonlinear SAE error.

# 7 REDUCING NONLINEARERROR($\mathbf{x}$)

In this section, we investigate whether simple techniques can reduce NonlinearError($\mathbf{x}$).

## 7.1 USING A MORE POWERFUL ENCODER

Our first approach for reducing nonlinear error is to try improving the encoder. We use a recent approach suggested by Smith (2024a): applying a greedy inference time optimization algorithm called gradient pursuit to a frozen learned SAE decoder matrix. We implement the algorithm exactly as described by Smith (2024a) and run it on all layer 20 Gemma Scope $9b$ SAEs closest to $L_0 \approx 60$. For each example $\boldsymbol{x}$ with reconstruction $\text{Sae}(\mathbf{x})(\boldsymbol{x})$, we use the gradient pursuit implementation with an $L_0$ exactly equal to the $L_0$ of $\boldsymbol{x}$ in the original $\text{Sae}(\mathbf{x})(\boldsymbol{x})$.

Using these new reconstructions of $\mathbf{x}$, we again run the test from Section 4.1 and do a linear transformation from $\mathbf{x}$ to the inference time optimized reconstructions. We then regenerate a similar scaling plot as Fig. 1 and show this figure in Fig. 8a. Our first finding is that pursuit indeed decreases the FVU of $\text{Sae}(\mathbf{x})$ by 3 to $5\%$; as Smith (2024a) only showed an improvement on a small 1 layer model, to the best of our knowledge we are the first to show this result on state of the art SAEs. Our most interesting finding, however, is that the variance explained by NonlinearError($\mathbf{x}$) stays almost constant when compared to the original SAE scaling in Fig. 1. In other words, if our tests are accurate, most of the reduction in FVU comes from better learning Dense($\mathbf{x}$) and reducing the linearly explainable error. Thus, in Fig. 1, we plot the additional reduction in NonlinearError($\mathbf{x}$) as the contribution of encoder error, and because NonlinearError($\mathbf{x}$) stays almost constant this section is very narrow.

## 7.2 LINEAR PROJECTIONS BETWEEN ADJACENT SAEs

Our second approach for reducing nonlinear error is to try to linearly explain it in terms of the outputs of previous SAEs. The motivation for this approach is that during circuit analysis (see e.g. Marks et al. (2024)), an SAE is trained for every component in the model, and being able to explain parts of the SAE error in terms of prior SAEs would directly decrease the magnitude of noise terms in the discovered SAE feature circuits. For the Gemma 2 architecture at the locations the SAEs are trained on, each residual activation can be decomposed in terms of prior components:

$$\text{Resid}_{\text{layer}} = \text{MLP\_out}_{\text{layer}} + \text{RMSNorm}(\text{O}_{\text{proj}}(\text{Attn\_out}_{\text{layer}})) + \text{Resid}_{\text{layer}-1} \qquad (5)$$

In Fig. 8b, we plot the $R^2$ of a regression from each of these right hand side components to each of the different components of an SAE trained on $\text{Resid}_{\text{layer}}$ (SaeError($\mathbf{x}$), LinearError($\mathbf{x}$), and NonlinearError($\mathbf{x}$)). We find that we can explain a small amount (up to $\approx 10\%$) of total SaeError($\mathbf{x}$) using previous components, which may be immediately useful for circuit analysis.

We also find that current layer's SAE reconstruction is able to explain $50\%$ of the variance in the nonlinear error, although this may not be entirely surprising, as the nonlinear error is a function of $\texttt{Sae}(\mathbf{x})$:

$$\texttt{NonlinearError}(\mathbf{x}) = \texttt{SaeError}(\mathbf{x}) - \texttt{LinearError}(\mathbf{x})$$
$$= (\mathbf{x} - \texttt{Sae}(\mathbf{x})) - \texttt{LinearError}(\mathbf{x})$$

These results mean that we might be able to explain some of the SAE Error using a circuits level view, but that overall, even in this setting, we will still have large parts of each error component unexplained.

## 8 CONCLUSION

The fact that SAE error can be predicted and analyzed at all is surprising; thus, our findings are intriguing evidence that SAE error, and not just SAE reconstructions, are worthy of analysis. Additionally, the presence of constant nonlinear error implies that current SAEs may have room for improvement, and therefore scaling SAEs may not be the only (or best) way to explain more of model behavior. The precise research direction to take to reduce nonlinear error depends on exactly why the nonlinear error arises. If it arises because the dictionaries SAEs currently learn are not good enough, improved SAE training procedures may suffice, while if the root cause is the sparsity constraint itself, future work might need to explore alternative simplicity penalties besides sparsity. We note that our tests are also approximate; we argue for the *existence* of $\texttt{NonlinearError}(\mathbf{x})$ as a separate term from $\texttt{Dense}(\mathbf{x})$, but the exact magnitude of each component remains uncertain. Ultimately, we believe that there is still room to make SAEs better, not just bigger.

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

## A  THEORY

Say we have a set of $m$ unit vectors $\boldsymbol{y}_1, \boldsymbol{y}_2, \ldots, \boldsymbol{y}_m \in \mathbb{R}^d$. We will call these "feature vectors". Define $\boldsymbol{Y} \in \mathbb{R}^{d \times m}$ as the matrix with the feature vectors as columns. We then define the Gram matrix $\mathbf{G_Y} \in \mathbb{R}^{m \times m}$ of dot products on $\mathbf{Y}$:

$$(\mathbf{G_Y})_{ij} = (\mathbf{Y}^T \mathbf{Y})_{ij} = \boldsymbol{y}_i \cdot \boldsymbol{y}_j$$

We now will define a random column vector $\mathbf{x}$ that is a weighted positive sum of the $m$ feature vectors, that is, $\mathbf{x} = \sum_i \mathrm{w}_i \boldsymbol{y}_i$ for a non-negative random vector $\mathbf{w} \in \mathbb{R}^m$. We say feature vector $\boldsymbol{y}_i$ is active if $\mathrm{w}_i > 0$. We now define the autocorrelation matrix $\mathbf{R_w} \in \mathbb{R}^{m \times m}$ for $\mathbf{w}$ as

$$\mathbf{R} = \mathbb{E}(\mathbf{w}\mathbf{w}^T).$$

We are interested in breaking down $\mathbf{x}$ into its components, so we define a random matrix $\mathbf{X}$ as $\mathbf{X}_{ij} = \mathrm{w}_j \mathbf{Y}_{ij}$, i.e. the columns of $\boldsymbol{Y}$ multiplied by $\mathbf{w}$. We can now define the Gram matrix $\mathbf{G_X} \in \mathbb{R}^{m \times m}$:

$$(\mathbf{G_X})_{ij} = (\mathbf{X}^T \mathbf{X})_{ij} = \mathrm{w}_i \mathrm{w}_j \boldsymbol{y}_i \cdot \boldsymbol{y}_j$$
$$\mathbf{G_X} = (\mathbf{w}\mathbf{w}^T) \odot \mathbf{G_Y}$$
$$\mathbb{E}(\mathbf{G_X}) = \mathbf{R_w} \odot \mathbf{G_Y},$$

where $\odot$ denotes Schur (elementwise) multiplication. The intuition here is that the expected dot product between columns of $\mathbf{X}$ depends on the dot product between the corresponding columns of $\mathbf{Y}$ and the correlation of the corresponding elements of the random vector.

We will now examine the L2 norm of $\mathbf{x}$:

$$\|\mathbf{x}\|_2^2 = \sum_{ij} \mathrm{w}_i \mathrm{w}_j \mathbf{y}_i \mathbf{y}_j$$
$$= \left\| (\mathbf{w}^T \mathbf{w}) \odot \mathbf{G_Y} \right\|_F^2 = \mathrm{Tr}(\mathbf{w}\mathbf{w}^t \mathbf{G_Y}) = \mathbf{w}\mathbf{G_Y}\mathbf{w}^T$$

We can also take the expected value:

$$\mathbb{E}(\|\mathbf{x}\|_2^2) = \mathrm{Tr}(\mathbf{R_w}\mathbf{G_Y})$$

Our goal is to find a direction $\boldsymbol{a} \in \mathbb{R}^d$ that when dotted with $\mathbf{x}$ predicts $\|\mathbf{x}\|_2^2$. In other words, we want to find $\mathbf{a}$ such that

$$\|\mathbf{x}\|_2^2 \approx \boldsymbol{a}^T \mathbf{x} = \boldsymbol{a}^T \sum_i \mathbf{w}_i \mathbf{y}_i = \boldsymbol{a}^T \mathbf{Y} \mathbf{w}$$

Combining equations, we want to find $\mathbf{a}$ such that

$$\boldsymbol{a}^T \mathbf{Y} \mathbf{w} \approx \|\mathbf{x}\|_2^2 = (vw^T \mathbf{G_Y} \mathbf{w})$$

Let us first consider the simple case where for all $i \neq j$, $y_i$ and $y_j$ are perpendicular. Then our goal is to find $\boldsymbol{a}$ such that

$$\boldsymbol{a}^T \mathbf{Y} \mathbf{w} \approx \text{Tr}(\mathbf{w} \mathbf{G_Y} \mathbf{w}^T) = \sum_i \langle y_i, y_i \rangle \mathbf{w}_i^2 = \sum_i \mathbf{w}_i^2 = \|\mathbf{w}\|_2^2 = \mathbf{w}^T \mathbf{w}$$

Since all of the $y_i$ are perpendicular, WLOG we can write $\boldsymbol{a} = \sum_i b_i \boldsymbol{y}_i + \boldsymbol{c}$ for a vector $\boldsymbol{c} \in \mathbb{R}^d$ perpendicular to all $\boldsymbol{y}_i$ and a vector $\boldsymbol{b} \in \mathbb{R}^m$. Then we have

$$\boldsymbol{a}^T \mathbf{Y} \mathbf{w} = \left( \sum_i b_i \boldsymbol{y}_i + \boldsymbol{c} \right)^T \mathbf{Y} \mathbf{w}$$
$$= \boldsymbol{b}^T \mathbf{w}$$

Since ordinary least squares produces an unbiased estimator, we know that if we use ordinary least squares to solve for $\boldsymbol{b}$, $\mathbb{E}(\boldsymbol{b}^T \mathbf{w}) = \mathbb{E}(\mathbf{w}^T \mathbf{w})$. Thus,

$$\sum_i b_i \mathbb{E}(w_i) = \sum_i \mathbb{E}(w_i^2)$$
$$b_i = \mathbb{E}(w_i^2)/\mathbb{E}(w_i)$$

Now that we have $b_i$, we can solve for the correlation coefficient between $\boldsymbol{a}^T \mathbf{x} = \boldsymbol{b}^T \mathbf{w}$ and $\|\mathbf{x}\|_2^2 = \mathbf{w}^T \mathbf{w}$. This gets messy when using general distributions, so we focus on a few simple cases.

The first is the case where each $\mathbf{w}_i$ is a scaled independent Bernoulli distribution, so $w_i$ is $s_i$ with probability $p_i$ and 0 otherwise. Then $b_i = s_i$. We also have that $\mathbb{E}(\mathbf{w}^T \mathbf{w}) = \mathbb{E}(\boldsymbol{b}^T \mathbf{w}) = \sum_i s_i^2 p_i = \mu$.

$$\rho = \frac{\mathbb{E}(\boldsymbol{b}^T \mathbf{w} \mathbf{w}^T \mathbf{w}) - \mu^2}{\sqrt{\mathbb{E}(\mathbf{w}^T \mathbf{w} \mathbf{w}^T \mathbf{w}) - \mu^2} \sqrt{\mathbb{E}(\boldsymbol{b}^T \mathbf{w} \boldsymbol{b}^T \mathbf{w}) - \mu^2}}$$
$$= \frac{\sum_i s_i^4 (p_i - p_i^2)}{\sqrt{\sum_i s_i^4 (p_i - p_i^2)} \sqrt{\sum_i s_i^4 (p_i - p_i^2)}} = 1$$

That is, for Bernoulli variables, $\mathbf{x} = \sum_i s_i \boldsymbol{y}_i$ is a perfect regression vector.

The second is the case when each $\mathbf{w}_i$ is an independent Poisson distribution with parameter $\lambda_i$. Then $\mathbb{E}(\mathbf{w}_i) = \lambda_i$ and $\mathbb{E}(\mathbf{w}_i^2) = \lambda_i^2 + \lambda_i$, so $b_i = \lambda_i + 1$. We also have that $\mathbb{E}(\mathbf{w}^T \mathbf{w}) = \mathbb{E}(\boldsymbol{b}^T \mathbf{w}) = \sum_i \lambda_i^2 + \lambda_i = \mu$. Finally, we will use the fact that $\mathbb{E}(\mathbf{w}_i^3) = \lambda_i^3 + 3\lambda_i^2 + \lambda_i$ and $\mathbb{E}(\mathbf{w}_i^4) = \lambda^4 + 6\lambda^3 + 7\lambda^2 + \lambda$. Then via algebra we have that

$$\rho = \frac{\sum_i 2\lambda_i^3 + 3\lambda_i^2 + \lambda_i}{\sqrt{\sum_i 4\lambda^3 + 6\lambda_i^3 + \lambda_i} \sqrt{\sum_i \lambda_i^3 + 2\lambda_i^2 + \lambda_i}}$$

For the special case $\lambda_i = 1$, we then have

$$\rho = \frac{6}{\sqrt{66}} \approx 0.73$$

## B  SYNTHETIC EXPERIMENTS WITH RANDOM DATA

For this set of experiments, we generated a random vector $\mathbf{x}'$ that was the sum of a power law of $100k$ random gaussian vectors in $\mathbb{R}^{4000}$ with expected $L_0$ of around 100. To simulate the SAE

reconstruction and SAE error, we simply masked a portion of the vectors in the sum of $\mathbf{x}'$. Unlike the more realistic synthetic data case we describe in Section 4.1, this did not work as expected: even in the case with no noise added to $\mathbf{x}'$ or the simulated reconstruction, the variance explained by the sum of the linear estimate of the error plus the reconstructed vectors plotted against the number of features "ablated" formed a parabola (with minimum variance explained in the middle region), as opposed to a straight line as in Fig. 2.

We note that this result is not entirely surprising: other works have found that random vectors are a bad synthetic test case for language model activations. For example, in the setting of model sensitivity to perturbations of activations, Giglemiani et al. (2024) found they needed to control for both sparsity and cosine similarity of SAE latents to produce synthetic vectors that mimic SAE latents when perturbed.

## C  NOTE ON FEATURE SHRINKAGE

Earlier SAE variants were prone to *feature shrinkage*: the observation that $\texttt{Sae}(\mathbf{x})$ systematically undershot $\mathbf{x}$. Current state of the art SAE variants (e.g. JumpReLU SAEs, which we examine in this work), are less vulnerable to this problem, although we still find that Gemma Scope reconstructions have about a 10% smaller norm than $\mathbf{x}$. One potential concern is that the $\mathbf{a}$ in Section 4.1 that we learn is merely predicting this shrinkage. If this was the case, then the cosine similarity of the linear error prediction $\boldsymbol{z}^T\mathbf{x}$ would be close to 1; however, in practice we find that it is around 0.5, so $\boldsymbol{z}$ is indeed doing more than predicting shrinkage.

