# OpenReview forum: "Decomposing The Dark Matter of Sparse Autoencoders"
_ICLR.cc/2025/Conference — ICLR 2025 Conference Withdrawn Submission_

### Official Review · Reviewer_MAo1 · 2024-10-27

**Soundness:** 2
**Presentation:** 2
**Contribution:** 3
**Rating:** 3
**Confidence:** 2

**Summary:**

This paper attempts to decompose the reconstruction error of sparse autoencoders into different interpetable components: (1) unlearned linear features, (2) residual dense features, and (3) nonlinear error introduced by the SAE. The authors explain their decomposition and then proceed to try to measure various components of the errors. They attempt to measure the linear and nonlinear portions of the error by training a linear regression on the activation vectors to predict the SAE recontruction error. They claim to find some irreducible nonlinear error arising from the SAE which does not disappear as the number of parameters in the SAE increases.

**Strengths:**

- The core idea to decompose the SAE reconstruction area into various parts is very interesting and, as far as I am aware, novel. Having such a decomposition could potentially be very useful for training better SAEs and informing the debate around the extent to which the linear representation hypothesis holds.
- The authors attempt to measure these quantities using a wide range of experiments. They also study the downstream effects of each component of their decomposition, and investigate methods for reducing the nonlinear error. Overall, I think the breadth of their empirical work is good.

**Weaknesses:**

The main weakness of this paper is that the central quantities are not defined clearly enough for me to be able to properly understand what is going on, or the adequacy of their empirical experiments. My current understanding, rephrased in my own language is that the authors are making the following decomposition:
- Activation vector $x$
    - The sparse autoencoder reconstruction: $\textup{Sae}(x)$
    - The error of the SAE reconstruction: $\textup{SaeError}(x)$
        - Unlearned sparse features: $\sum_{i=m}^n w_i y_i$
        - Dense feaures which will not be learned by the SAE: $\textup{Dense}(x)$.
        - Additional error: (I don't know if the authors have a term for this)
            - Linearly predictable additional error: $Wx$
            - Nonlinear error: $\textup{NonlinearError}(x)$
(where my notation is that each bullet point is the sum of the bullet points nested at one level below it). This was my understanding after having spent quite some time re-reading Section 3, so I'll proceed as if this is correct, but I'm not confident and I'd appreciate clarification from the authors.

Assuming the above, I have several comments:
- The exposition in Section 3 needs to be substantially clearer for readers to be able to understand your definitions. For example, you should name the error mentioned in "the SAE introduces some error when making this approximation" and give it a consistent name. I also think equations (3) and (4) are misleading. According to my understanding, (4) is the central definition decomposing the SAE error and (3) is a consequence of (4) when you consider subtracting from $x$.
    - As an aside, the introduction of $W$ is poorly motivated here. Also, later on you're going to consider another sense in which the error can be linearly predicted from $x$ - namely using $a^T x$ to predict $\textup{SaeError}(x)$. I believe that these are not the same, but I was initially confused here and I don't understand the difference in intuition between them.
- In Figure 1, I don't see why Dense Features and Linear Error should be grouped together. They seem like quite different quantities with different interpretations to me.
- The changes of notation and setup in the first paragraph on page 6 are very unhelpful given there's already a lack of clarity. I'd strongly recommend that the authors stick to consistent notation throughout the paper, and choose a single - clearly distinct - term and corresponding notation for each component of their decomposition and stick with it throughout the paper.
    - As an aside, if we're dropping $Wx$ in this paragraph, why did we have it in the first place? I'm not sure I ever understood the intuition for having it.
- Since you are agnostic to the SAE architecture, you don't introduce any notation for the features that the SAE learns. This lead to me getting confused and originally not understanding that the SAE reconstruction error comes from essentially four places: learning wrong features, learning wrong feature weights, unlearned linear features, and (unlearned) dense features. My understanding is that the nonlinear error is effectively measuring the first two. Is that right? If so, a clarifying comment in this direction might be helpful.

Secondly, I did not understand the description of and intuition behind the experiment in the first paragraph of page 4. I think this might be downstream of the fact that I haven't completely understood the authors' decomposition of the SAE error. But in any case, either the authors need to offer a clearer definition of the quantities they are working with, or more intuition for why this experiment claims to measure what they are hoping - and probably both.

Given this lack of clarity in a couple of key places in the manuscript, it was hard for me to engage with the more detailed experimental results in Sections 5 onwards, since I couldn't understand what the authors were actually hoping to measure. The problem that the authors are trying to study seems fundamentally very interesting, and I'm optimistic that some version of this paper could be very solid, but as it stands it's not possible to appreciate the authors' contributions.

More minor points:
- I think the synthetic experiment setup in Section 4 is unlikely to be particularly realistic - particularly the assumption of Gaussian noise. (My read of Gurnee (2024) which the authors cite regarding pathological reconstruction errors makes the Gaussian assumption likely incorrect.) But, this is not a central concern.
- I suggest that the authors read over their manuscript for minor typographical errors. Ones that I caught include: issues with citation formatting in several places, summation indices in equations (3) and (4) being incorrect, and several stray commas.

**Questions:**

Combined with previous section for clarity.

---

### Official Review · Reviewer_zGqo · 2024-10-31

**Soundness:** 2
**Presentation:** 1
**Contribution:** 2
**Rating:** 3
**Confidence:** 2

**Summary:**

This paper tries to decompose the error components of sparse autoencoders (SAEs) to help better interpret language models. It uncovers that the SAE error can be predicted and analyzed, and provides insights for reducing it.

**Strengths:**

1. This paper focuses on the commonly ignored detail—SAE error—to provide more insights for thoroughly understanding a research subject related to the representation of language.
2. It examines SAE error from multiple aspects, such as scaling law, norm prediction test, etc.
3. It investigates ways to reduce NonlinearError in detail.

**Weaknesses:**

# Major

To be honest, I had a very hard time understanding this paper. Specifically,

1. How do you define the term “most common” (features..) on line 146? It is ambiguous.
2. Section 4.1 is not well written, which makes me hard to understand the later sections of the paper. Specifically:
    1. Line 193 and 194 do not make sense. For instance, Dense is nonlinear, how can the sum between Wx and Dense be linear component of the error?
    2. What leads you to make the statement on line 200 to 201? You don’t even know what the “true features” are. Please elucidate.
    3. It is questionable to claim “the percent of variance left unexplained by the regression will be an upper bound on the true variance explained by NonlinearError(x).” First, please provide a clear definition of “variance explained or unexplained”, perhaps in the appendix. Second, if this is a concept similar to the explained variance in PCA, I’m not sure if you can readily extrapolate that to nonlinear components. Some rigorous derivation is needed. If you are not sure, you need to stress this is an assumption.
    4. Here the so called synthetic setup is just for confirming that the *SAE is approximately an identity function on its image, i.e., the linear subspace of* $\mathbf{x}'$. There is no need for this long verbosity.
        1. Also, what is the motivation for this? I think there should be more clarification.
    5. Linear transformation $\mathbf{a}$ (line 161) should be denoted by $A$ for consistency (since you have $W\mathbf{x}$). You current notation makes it look like evaluating inner product.  I don’t think this is the same as the one in Section 4.2, right? Also, clarify its output dimension.
    6. The Gaussian noise simulation on line 204 to 207 is kinda questionable.
        1. The set of $\mathbf{x}$ could be a curved low dimensional manifold. In this case the Gaussian noise—whose support is a practically a ball of the same dimension as $X$—cannot accurately simulate Dense(x).
        2. The output of Dense and NonlinearError could be correlated in reality.
        3. The functions Dense and NonlinearError could be continuous, so Gaussian noise may not accurately simulate them.
    7. Since SAE is approximately an identity function according to you (line 204 to 205), we can safely assume that $\mathbf{x}’$ and SAE($\mathbf{x}’$) are identical. Then adding Gaussian noises to each of them simply makes the simulated SaeError (i.e., $\mathbf{x}$ - SAE($\mathbf{x}$)) an Gaussian noise. I’m not sure how using a linear map $\mathbf{a}^\top \mathbf{x}$ can derive the results in Fig. 2, or maybe I just got it completely wrong since I can hardly understand your writing. I think you might need to elaborate on line 202 to 207.

I’d like to stop here since Section 4.1 has already baffled me enough, making me impossible to review the later sections. I think some revision is needed to streamline the narrative.

# Minor

1. On line 135, I think you mean $\|\mathbf{w}\|_0\ll d$.
2. What is called the “linear subspace of $\mathbf{x}$” (line 187)? $\mathbf{x}$ is just a point in the activation space $X\ni\mathbf{x}$. If you mean a linear subspace of $X$, then is it *proper*, i.e., it cannot be $X$ itself?
3. Use the correct citation format \citep and \citet. For instance, line 35.

**Questions:**

Check the weaknesses.

---

### Official Review · Reviewer_un5Q · 2024-11-04

**Soundness:** 2
**Presentation:** 1
**Contribution:** 3
**Rating:** 5
**Confidence:** 2

**Summary:**

This paper analyzes sparse autoencoder (SAE) errors in language models and found they could decompose these errors into three parts: unlearned sparse linear features, a dense linear term and a "nonlinear error" term that persists even as SAEs get larger. The paper studies the nonlinear error across different token posistions and SAE widths, with experiments on Gemma-2 9B. The paper attempts to reduce this nonlinear error through two methods: using gradient pursuit during inference (which only slightly helped) and leveraging SAE reconstructions from adjacent model components.

I found this paper complex, and while I think it is flawed in the current state I am happy to revise my opinion if my concerns are addressed.

**Strengths:**

* I think that the biggest two problems with Sparse Autoencoders and Mechanistic Intepretability research is i) faithfulness of decomposition of models and ii) real-world application of insights. This paper is solid progress on i) because they ask why SAEs are currently limited.

* The paper describes a sensible theoretical decomposition of model activations into SAE learned features, dense features and non-linear features, and also measures these terms in practice.

* The paper does a wide range of analyses: automated interpretability, FVU and loss measurements.

**Weaknesses:**

1. **Large sections of the paper are difficult to understand**

a. Some notation is defined in strange ways that require me to reread them many times. E.g. `SaeError(x) := x - Sae(x)` is defined with a -1 coefficient for `Sae(x)`, but `NonlinearError(x) := Sae(x) - Wx  - \sum_{i=0}^{m} w_i \vec{y}_i` is defined with a +1 coefficient for `Sae(x)`. I *think* that most of these problems are downstream of defining some parts of the notation with an end state in mind, e.g. the weak linear representation hypothesis suggests that there exists sum ideal dictionary decomposition of the vector, but other parts of the notation is defined with the current state in mind, e.g. you slot `SaeError(x)` into the sum with the existing `Sae(x)` and `x` terms and nothing else. However, I'm not confident that this consistently considering solely the end state or current state is either necessary or sufficient for making the notation clearer.

b. Some statements do not make sense after several times re-reading. E.g. "The intuition behind this test is that if ... its existence is not guaranteed".

c. "If this test is accurate, we can use it to estimate the linear component of the error, `Wx + Dense(x)`" but `Dense(x)` was introduced as possibly non-linear, and `Wx` is non-linear, so how is this the **linear** part of the error?

2. **I think the conclusions are too strong**.

The paper states "We also find that the norm of the `NonlinearError(x)` is constant on a per token level as
we scale SAE width", and while this section changes the definition of `NonlinearError` to be the error as determined by their method, in the conclusion the paper states "... the presence of constant nonlinear error ..." with no hedging.

I am not convinced that the methods in the paper are capturing *true* linear error and non-linear error. One reason this may be happening is that all the training on the error term `x - Sae(x)` may involve some vestigual `x` due to shrinkage (which could be boosted by 10x by the various predictors, since the appendix suggests there is 10% shrinkage). This would mean the various methods that predict linear or nonlinear error may be cheating and picking up on this shrinkage. Is this addressed in the paper? Why is there no hedging in the conclusion that the methods presented may not capture true (non-)linear errors, or even that the weak linear representation hypothesis may not be true. For example, in the extreme case where the SAE was entirely dead, we can predict `SaeError(x) = I * x` and so the SAE solely has linear error, which suggests there are some assumptions that need to be stated (that the SAE is sufficiently good at reconstructing I think).

**Questions:**

1. Why is it necessary to sanity check that estimated linear errors and estimated non-linear errors are correlated in Section 4.1? I found this experiment technical and in-the-weeds and couldn't tell why it needed be done (the interesting parts of the paper are with the actual LLM and actual error).

2. Why are all the linear extractors single linear matrices, or another SAE? Shouldn't an SAE with no sparsity penalty be applied somewhere so it's possible to learn features in superposition without interference (not possible with a plain linear matrix) and features that are dense (not possible with a sparsity penalty)?

---

### Official Review · Reviewer_SuAh · 2024-11-08

**Soundness:** 2
**Presentation:** 4
**Contribution:** 3
**Rating:** 3
**Confidence:** 4

**Summary:**

This paper is an analysis into the error of sparse autoencoders applied to LLM interpredability. They address the shortcoming of the ability to reconstruct the hidden state, and observe that scaling laws show that an SAE would not be able to fully represent the hidden state, even in the limit. The theoretical framework is applied to decomposing the error of one layer of the Gemma 2 open source LLM, using the open source SAE published in a separate work, which agrees with the breakdown.

**Strengths:**

The paper’s analysis is very thorough and a very compelling decomposition of the scaling of sparse autoencoders for feature extraction. The observations are very well described, and the theory and methodology is sound. I think it would be an impactful work, however, it does have one fatal shortcoming: (see below)

**Weaknesses:**

Weaknesses
The main weakness is that the method is only applied in one model, and even worse, only one layer in the model. The methodology is self described as empirical, but because only one context is analyzed, the findings are not sufficiently proven because the paper essentially shows only one data point. That is, is it possible that this only happens in this one layer for this one model? I would not think that the findings are exclusive to just this one context, but would it be possible to construct a LLM architecture for which this result does not hold? It should be easy to repeat the analysis for various open source models and different layers, and to thus illustrate that the observations and theory hold more widely. With just more demonstrations of the same trends across more models, the paper would be strong.


Another easy to correct shortcoming in the presentation is that details of the autoencoder training process are left out. There is a citation to GemmaScope, but given that this paper is trying to make a broader theoretical claim, a short description of the SAE training process would be apt. Similar to applying the method to more models and layers, it would also be interesting to verify if different SAE algorithms resulted in features that change the results.

**Questions:**

- Why was Gemma analyzed? Why was Gemma Scope chosen? What is special about these choices? (Availability as an open source model with a published SAE is an acceptable answer, but it that should be stated in the paper that the choice was made out of convenience.)
- Did you train the SAE from scratch, or are you using a the published checkpoint?
- What type of sparse autoencoder is considered? How was it trained? (As mentioned above, briefly describe the technique if you are using a published checkpoint.)
- Figure 1:
  - What are the equations in the labels supposed to mean?
  - Where did the data come from?
  - Exactly which part of the figure is “dark matter”?
- Equation 1:
  - is the vector w “random” or is it actually a coordinate of x in the y basis?
  - Is ||w||_1 << d the right equation? Couldn’t one of the components be much larger than one?
- Line 196: What is the set up of the synthetic set up?
- What is L_0?
- Line 242 --The exact set up of the random vectors here was unclear.
- What is Figure 3b supposed to show?
- Line 240: Exactly what is “likely” supposed to mean?
- Line 264: Bibliography: Gemma reference has a broken last name: “Team”
- Figure 4: The caption is not a full sentence and not clear. Expand the caption.
- What does the last equation on Line 490 say?
- Appendix A:
  - This section needs a little more introduction: What is the proof trying to show, and what does it imply?
  - Line 697: Why is the case \lambda=1 special, and what do we conclude from rho=0.73 ?
  - Is d > m?
  - Line 662: define WLOG
- Appendix B: report the results in the appendix, instead of just qualitatively describing them.
- Does Appendix C have a reference in the main text?

---

### Note · Authors · 2024-11-19

**Comment:**

We greatly appreciate the reviewers comments and suggestions!

It seems to us that the reviewers all agree that we have interesting findings, but that we need to clarify many parts of the paper and perhaps run a broader set of experiments. Thus, we have decided to withdraw our work and improve it in these aspects.

Thank you all again so much for your time and expertise.

**Withdrawal Confirmation:**

I have read and agree with the venue's withdrawal policy on behalf of myself and my co-authors.